# Biology of Peptide Transporter 2 in Mammals: New Insights into Its Function, Structure and Regulation

**DOI:** 10.3390/cells11182874

**Published:** 2022-09-14

**Authors:** Caihong Wang, Chu Chu, Xiang Ji, Guoliang Luo, Chunling Xu, Houhong He, Jianbiao Yao, Jian Wu, Jiangning Hu, Yuanxiang Jin

**Affiliations:** 1College of Biotechnology and Bioengineering, Zhejiang University of Technology, Hangzhou 310032, China; 2Department of Bioinformatics, College of Life Sciences, Zhejiang University, Hangzhou 310058, China; 3Zhejiang Conba Pharmaceutical Limited Company, Hangzhou 310052, China; 4Zhejiang Institute of Modern Chinese Medicine and Natural Medicine, Hangzhou 310052, China

**Keywords:** PepT2, membrane transporter, proton-coupled oligopeptide transporter, peptide

## Abstract

Peptide transporter 2 (PepT2) in mammals plays essential roles in the reabsorption and conservation of peptide-bound amino acids in the kidney and in maintaining neuropeptide homeostasis in the brain. It is also of significant medical and pharmacological significance in the absorption and disposing of peptide-like drugs, including angiotensin-converting enzyme inhibitors, β-lactam antibiotics and antiviral prodrugs. Understanding the structure, function and regulation of PepT2 is of emerging interest in nutrition, medical and pharmacological research. In this review, we provide a comprehensive overview of the structure, substrate preferences and localization of PepT2 in mammals. As PepT2 is expressed in various organs, its function in the liver, kidney, brain, heart, lung and mammary gland has also been addressed. Finally, the regulatory factors that affect the expression and function of PepT2, such as transcriptional activation and posttranslational modification, are also discussed.

## 1. Introduction

Mammalian peptide transport is a crucial way to obtain nitrogen for many organisms in the form of dipeptides and tripeptides from ingested proteins and reserve peptides via reabsorption in the kidneys [1]. In mammals, the proton-coupled oligopeptide transporter (POT) family mediates the absorption of dipeptides and tripeptides into cells via an inwardly directed proton electrochemical potential gradient [2,3]. The POT family in eukaryotes contains four members, including peptide transporter 1 (PepT1), peptide transporter 2 (PepT2), peptide histidine transporter 1 (PhT1) and peptide histidine transporter 2 (PhT2). PepT1 and PepT2 are able to mediate the absorption of almost all di- and tripeptides and many peptidomimetics [4], such as β-lactam antibiotics [5], 5-aminolevulinic acid [6] and antiviral agents [7]. Histidine, di- and tripeptides are transported by PhT1 and PhT2 into the cell [2,3].

PepT2, a high-affinity, low-capacity transporter, has a higher affinity for the same substrates than PepT1 [8]. PepT2 is mainly located in the kidney, brain and lung [9]. In mammals, PepT2 contributes to the conservation of amino acid nitrogen and drug disposition [10]. Almost all dipeptides, tripeptides and peptidomimetic drugs, including β-lactam antibiotics and angiotensin-converting enzyme inhibitors, can be absorbed by PepT2 [11]. PepT2 has recently attracted extensive attention due to its application in drug delivery [12,13,14].

Therefore, improving our understanding of the expression, regulation and transport mechanism of PepT2 in mammals is an important challenge in physiology and pharmacology. This review addresses the current knowledge of PepT2, as well as its regulation and physiological and pharmacokinetic roles.

## 2. Tissue Expression of PepT2

The physiological, pharmacological and pathological functions of PepT2 in mammals depend on its expression level and cellular localization. Lu et al. (2006) examined the distribution of PepT2 in 19 tissues of rats and mice and found that PepT2 was mainly expressed in the kidney, lung and brain [9]. This study was the first report that revealed the expression of PepT2 in the pituitary and reproductive organs [9].

PepT2 is widely distributed in various tissues. In the kidney, it is predominantly expressed in the latter sections of the proximal tubule [15]. As PepT1 were found in S1 and other convoluted segments of the proximal tubule, so the distribution of peptide transporters is heterogeneous in kidney. Peptides and peptidomimetic drugs are absorbed sequentially in kidney, first by PepT1 and second by PepT2. It is also localized in brush border membranes of proximal tubule S3 segments and widely expressed in the brain, including the olfactory bulb, cerebral cortex, basal ganglia, hindbrain and cerebellum [12,16,17]. The choroid plexus is the location of the blood–cerebrospinal fluid barrier, and PepT2 has been confirmed to be expressed at this barrier [17,18,19,20,21]. In the rat lung, it is expressed in the bronchial epithelium, alveolar type II pneumocytes and the endothelium of small vessels [22]. In addition, it is also expressed in the mammary glands of humans, bovines, pigs and rabbits [23,24,25,26,27]. In the eye, PepT2 mRNA has been found to be localized in the retina of rats by in situ hybridization studies [28]. It is localized in keratinocytes, functions in skin oligopeptide uptake and is expressed in glial cells and the macrophages of the enteric nervous system [29,30]. PepT2 was also found to be expressed in splenic lymphocytes, macrophages, monocytes, granulocytes, erythrocytes and their committed precursors and it also has a much higher expression in mouse macrophages than in lymphocytes [31,32]. Table 1 summarizes the tissue distribution of PepT2 in mammals. Wherever it is expressed, PepT2 is found to be localized to the apical membrane of polarized cells [21,33,34,35].

## 3. Function of PepT2

PepT2 mediates the reabsorption of dipeptides and tripeptides and plays an essential role in the maintenance of mammalian protein nutrition [45,46,47]. Peptidomimetic drugs, including amino cephalosporins, peptidase inhibitors, angiotensin-converting enzyme inhibitors and many newly developed drugs, are substrates of PepT2 [4,5,6,7]. Hence, PepT2 is regarded as both a nutritional transporter and a drug transporter. The renal proximal tubule is the primary location of the reabsorption of the peptide [48]. The percentage of di- and tripeptides in the circulating plasma is approximately half of all amino acids [49]. PepT2 contributes to peptide reabsorption and amino acid homeostasis in tubular cells [50]. The reabsorption of dipeptides and tripeptides by PepT2 in the kidney is important for equilibrating protein intake according to the needs of the body. PepT2 knockout mice showed inhibited renal dipeptide accumulation, indicating that PepT2 may be necessary for the tubular reabsorption of dipeptides [39]. PepT2 can also play an essential role in retaining drugs and restricting their excretion in the urine and function in pharmacokinetics, tissue distribution and the renal handling of drugs [51]. Studies revealed that PepT2 was the sole transporter for the reabsorption of cefadroxil in the kidney [52]. Xu et al. (2014) revealed that PepT2 mediated the reabsorption of entecavir and reabsorbed approximately 25% of urinary entecavir [53]. Drug reabsorption via PepT2 in the kidney contributes to the maintenance of their systemic exposure.

In the brain, PepT2 functions in the absorption of peptides and peptidomimetics [13]. PepT2 localized in choroid plexus epithelial cells was proven to maintain neuropeptide homeostasis and the removal of neurotoxins from the brain [21,34,37,54]. Although PepT2-deficient mice exhibit no obvious kidney or brain abnormalities, they appeared reduced dipeptide uptake by the choroid plexus, indicating that PepT2 is responsible for the peptide transport through the blood–cerebrospinal fluid barrier [17]. The reason that PepT2-deficient mice exhibited no obvious kidney or brain abnormalities is that PepT2 is suspected to act as a redundant nitrogen and essential AA sources [55]. PepT2 can also have a pharmacological or toxic influence by affecting substance disposition. PepT2 null mice exhibit higher cerebrospinal fluid substrate concentrations and lower choroid plexus accumulations of GlySar, cefadroxil and 5-aminolevulinic acid (5-ALA) than wild-type mice [36,52,56]. Higher antinociceptive response appeared after L-kyotorphin administration in PepT2 null mice [56].

Ocheltree et al. (2004) characterized the functional significance of PepT2 at the blood–cerebrospinal fluid barrier and found that it is essential for the removal of di- and tripeptides and endogenous peptidomimetic substrates from cerebrospinal fluid [34]. They proved that only PepT2 is expressed in the choroid plexus [34]. PepT2 in brain cells also acts in neuropeptides homeostasis and peptidomimetic drugs distribution. PepT2 null mice exhibited lower neuropeptide absorption, including carnosine, L-kyotorphin and 5-aminolevulinic acid, than neonatal astrocytes from the brain parenchyma of wild-type mice [57,58,59]. Chen et al. (2017) proved that PepT2 is involved in the distribution of cefadroxil in brain parenchyma, brain extracellular fluid and cerebrospinal fluid and convincingly demonstrated that PepT2 removes peptidomimetic drugs from the cerebrospinal fluid to brain cells [12]. In brief, all of these studies suggest that PepT2 is essential in regulating the physiological, pharmacological and toxicological effects of small peptides and peptidomimetic drugs in the brain.

The presence of PepT2 in the lung is important for developing new strategies to deliver drugs for treating infectious and neoplastic diseases [22,60]. PepT2 was expressed in the bronchial epithelium, alveolar type II pneumocytes and endothelium of small arteries of rat lung. It is responsible for peptide and peptidomimetic transportation in pulmonary epithelial cells. In conclusion, PepT2 is expected to be responsible for peptide and peptidomimetic delivery to treat pulmonary disease pulmonary disease [61]. Physiologically, PepT2 can take up bacterial dipeptides, such as γ-D-glutamyl-meso-diaminopimelic acid (γ-iE-DAP), resulting in an innate immune response [62].

PepT2 is mainly located in the kidney, brain and lung, and recently it has also been found to be located in mammary gland, neurons and keratinocytes. In the mammary gland, PepT2 contributes to the reuptake of small peptides derived from milk protein hydrolysis in the lumen [22]. The uptake of some essential amino acids in the mammary gland is insufficient for milk protein synthesis [63]. Peptides in the blood could be taken up by PepT2 and used for milk protein synthesis in the mammary gland [15,41]. Wang et al. (2019) revealed that PepT2 is the major transporter involved in peptide uptake in bovine mammary epithelial cells (BMECs) [26]. Furthermore, Kudo et al. (2021) revealed that PepT2 is localized in keratinocytes and functions in skin oligopeptide uptake [29]. Lopachev et al. (2021) found the presence of PepT2 in neurons, and the PepT2-dependent transport of carnosine in neurons is one of its main routes of uptake [64].

The function of PepT2 in the kidney, mammary gland, lung and choroid plexus has been proven, and PepT2 has been found to be responsible for peptide transport and drug delivery in these tissues. As PepT2 transports various peptidomimetics and controls the pharmacokinetics and pharmacodynamics of these drugs, it can influence the pharmacological effects of peptidomimetic drugs [65]. Prodrugs targeting PepT2 show favorable drug absorption and have been extensively used in the medicine and biology fields [6,66,67]. Overall, PepT2 plays a necessary role in nutrition and pharmacology in mammals.

## 4. Structure of PepT2

The characterization of the structure of PepT2 is critical for understanding the essential determinants of its substrate binding and the transport mechanism. Human PepT2 (hPepT2) contains 729 amino acid residues involving 12 transmembrane domains (TMDs) (Figure 1) [10]. It contains a big extracellular loop between TMDs 9 and 10 [65,68]. Fei et al. (1998) identified a putative substrate binding site located at TMDs 7, 8 and 9 of these transporters by constructing PEPT1–PEPT2 and PEPT2–PEPT1 chimeras [68]. By using site-directed mutagenesis techniques, Arg^57^, His^121^, Tyr^56^, Tyr^64^ and Tyr^167^ are found to be responsible for substrate transport by PepT2 [69].

The rat PepT2 (rPepT2) cDNA contains 3938 bp, encoding a 729-amino acid protein with a poly(A)+ tail [70,71]. A great deal of research has been carried out to study the structure of PepT2. The topological structure of PepT2 was analyzed by site-directed mutagenesis studies, and the crucial amino acids for substrate binding were found [27]. In addition, Wang et al. revealed that bovine PepT2 (bPepT2) was predicted to have 12 TMDs, with a large extracellular loop between the TMDs 9 and 10 [27].

Structurally, PepT2 belongs to the major facilitator superfamily (MFS) that is usually composed of 12, but sometimes 14, trans-membrane (TM) helices [72]. PepT2 contains 12 TM helices, which adopt into the overall fold the MFS transporters LacY, GlpT and EmrD. Recently, structural information on mammalian PepT2 has been identified [73,74]. Killer et al. (2021) reported the cryo-electron microscopy (cryo-EM) structures of hPepT2 in a substrate-free state and in complex with Ala-Phe [73]. The cryo-EM structure revealed that hPepT2 and rPepT2 adopts the canonical fold of the MFS in an “outward-open” structure [73,74]. The region containing His^61^ was identified to be the primary protonation site in hPepT2 [75]. The structure of PepT2 in different species is conserved in mammals [73,74].

## 5. Substrates of PepT2

PepT2 has broad substrate specificity and absorbs small tripeptides with distinct sizes and charges [76]. Amino acids and tetrapeptides cannot be transported by PepT2 [51]. It is one of the most promiscuous transporters and can transport almost 400 dipeptides and 8000 tripeptides [77,78]. Although PepT2 can mediate the transportation of all di- and tripeptides, they have strikingly different affinities [79]. Peptides containing L-amino acids have a higher affinity than those consisting of D-amino acid residues, but peptides containing D-stereoisomers are not absorbed [80]. Substrates transported by PepT2 with high affinity have the following characteristics: (1) di- and tripeptides; (2) a acid amide bond or ketomethylene; (3) dipeptides with intramolecular distances between oppositely charged NH2 and COOH head groups >500 and <630 picometers and in zwitterionic forms; (4) a properly positioned backbone carbonyl group; (5) a free amino group in the α or β position; (6) stereoselectivity with L-amino acids and transconformers; (7) chiral centers on the α-carbons and backbone torsion angles ψ, φ and ω; (8) an infirmly basic group in the α-position at the N-terminus; and (9) an acidic or hydrophobic feature at the C-terminus [81,82,83,84].

Pharmacologically active peptidomimetic drugs comprising β-lactam antibiotics, valacyclovir and ACE inhibitors are recognized and transported by PepT2 [85,86,87,88,89]. The aminocephalosporin cephalexin was first identified to be transported via PepT2 in rat kidneys [90,91]. Polypeptide antibiotics, such as actinomycin D, colistin (polymyxin E) and polymyxin B, are all substrates of PepT2 [5]. PepT2 can regulate the spatial distribution of peptide-like drugs, including cefadroxil, in the brain [92]. As more PepT2-mediated drugs are discovered, they will become important targets for manipulating the delivery of drugs to specific organs and regulating the kinetics and dynamics of drugs [93].

## 6. Transport Mechanism of PepT2

The high level of the sequence of PepT2 between prokaryotes and eukaryotes indicates that they may work through a conserved transport mechanism [80]. The hPepT2 is formed by 12 TMDs with TM1 to TM6 constituting the N-terminal bundle and TM7 to TM12 constituting the C-terminal bundle, and N- and C-termini facing the cytoplasm [94]. The bundles are linked by a bundle bridge, which contains two helices interacting with each other. The peptide transport mediated by PepT2 is accomplished by the motility of the gating helices around the binding site [95]. The extracellular gate, which is composed of TM1 and 2 from the N-terminal bundle pack against TM7 and 8 from the C-terminal bundle, regulates extracellular peptide come close to the binding site. The intracellular gate, composed of TM4 and 5 stacking against TM10 and 11, regulates intracellular peptide and protons release. The interactions of two salt-bridges were found to orchestrate these helices and regulate the conformational state of PepT2 [96]. The intracellular gate involves a conserved lysine and glutamate pair, but the extracellular gate is not very conserved [97]. Mammalian PepT2, a conserved histidine on TM2 and an aspartate–arginine salt bridge on TM1 and 7 is identified [75]. The proton binding promotes the reorientation of PepT2 from the inward-open to outward-facing state, consequently affecting the reorientation step [98].

Recently, it was suggested that peptide transport proceeds through the “rocker-switch” alternating access mechanism, which contains conformational transitions of three diverse states: (i) outward open, (ii) occluded and (iii) inward open [99]. Lastly, the transport mechanism of mammalian PepT2 has been reported by Parker and his colleagues [74]. For rPepT2 (Figure 2), when it is in the outward-facing condition, it exhibits a great polar pore on the extracellular side of the membrane [74]. The histidine on TM2, His^87^, approaches the extracellular solution and is protonated (i). Once the N-terminus of di-/tripeptides are attracted to the Asp^317^ on TM7, di-/tripeptides come into the rPepT2. Then, the proton retained first on His^87^ moves to Asp^317^ before transferring again to Glu^622^. Under the impact of the dipole between the N- and C-terminal bundles of rPepT2, the peptide orientates into the binding site of the transporter with the C terminus of the peptide interrelating with Arg^57^, part of the conserved ExxER57 motif on TM1 (ii). Subsequently, TM1 to TM2 will shift to TM7 to TM8, close the extracellular portal and stabilize through the interaction between His^87^ and Asp^317^. When rPepT2 is in the occluded condition, the proton on His^87^ will transfer from Asp^317^ to Glu^622^ (iii). The binding of the proton with Glu^622^ will destroy the interaction between PepT2 with the N-terminus of the peptide. At the other end of the binding site, the ExxER57 motif protonation results in the destruction of its interactions with Lys127, making TM4 and TM5 vibrate to deliver the peptide and protons to the cytoplasm (iv). In the inward-open condition, a water-filled channel forming from the extracellular side of PepT2 to His^87^ results in the reprotonation of this side chain, disrupting its interrelation with TM7 and launching the reorientation of rPepT2 to the outward-facing condition, achieving a full circle [74].

The binding site of PepT2 is constituted by residues from helices H1, H2, H4 and H5 and from helices H7, H8, H10 and H11. Three conserved positively charged residues including Arg25, Arg32 and Lys127 are located on the N-terminal side of the binding site [100]. Studies revealed that mutation of Arg25 in hPepT2 completely inhibits its transport activity [69].

## 7. Regulation of PepT2

Given the importance of PepT2 for nutrient uptake and drug transport, the molecular regulation of PepT2 expression under various conditions needs to be understood. The expression and transport activity of PepT2 are controlled by different factors, including substrates, physiological status, development, age, pathological conditions and hormones [51]. The factors regulating PepT2 are presented in Table 2.

### 7.1. Genetic and Transcriptional Regulation of PepT2

It is widely accepted that the most feasible factors regulating transporters are their substrates. This is also true for PepT2 [101,111,112]. Many studies have examined this possibility and provided firm evidence in support of it. A study revealed that PepT2 was increased at the translational level by Met-Met treatment of BMECs [101]. Consistently, a subsequent study demonstrated dietary regulation of PepT2 [112]. Sun et al. (2020) demonstrated that protein restriction decreased the mRNA abundance of PepT2 in the kidneys of nursery pigs [112]. In addition, starvation reduces human renal peptide transport activity, as revealed by a decrease in the removal of intravenously infused Gly-Gln in the kidney [111].

### 7.2. Regulation by Hormones

Hormones, produced by the endocrine system, are signaling molecules that function as regulators of cell metabolism. The effects of selected hormones on PepT2 have been proven in the last decade [102]. For instance, the regulation of PepT2 by other lactogenic hormones, including prolactin, insulin and hydrocortisone, was also observed in BMECs [102]. In addition, epidermal growth factor (EGF) provoked a dose-dependent decrease in the transport capacity and expression of PepT2 in SKPT cells, as determined by the studies of the apical uptake of (14C) glycylsarcosine and its mRNA levels. These findings indicate that EGF inhibits PepT2 expression by reducing the transcription and mRNA stability of rat PepT2 in proximal tubule cell line SKPT [35]. However, Sondergaard et al. (2008) revealed that EGF increased the uptake of ^14^C-glycylsarcosine, without changing pig PepT2 (pPepT2) expression in the renal porcine LLC-PK1 cell line. The reason for this may be that each cell treated by EGF kept the same number of pPepT2 per cell as the non-treated cells [103]. An enhanced number of cells in the EGF-treated monolayers led to larger quantities of transporters per monolayer and enhanced the transport activity of pPepT2 [103]. Bravo et al. (2008) revealed that 8-day treatment with epidermal growth factor (EGF) enhanced peptide transport activity of PepT2 and resulted in upregulated transepithelial electrical resistance, total cell protein and alkaline phosphatase activity in LLC-PK1 cells [103]. This different effect of EGF on the expression and function of PepT2 may be due to the species. Lu et al. revealed that thyroidectomy upregulated the mRNA expression of PepT2 in male rat kidneys, and the increases were suppressed by thyroid hormone replacement [9]. Corticosteroids inhibit the function of PepT2 and result in lower bacterial peptide-induced alveolar epithelial cells to inhibit the innate immune response [104].

### 7.3. Regulation by Development and Age

The expression of PepT2 is regulated by the physiological state of the animal [23,24]. It has been revealed that the expression of PepT2 is modulated by development and age [23,24].

In the heart, PepT2 expression in Wistar rats was regulated by aging [113]. The expression of PepT2 in the hearts of middle-aged and old Wistar rats was significantly upregulated compared with that in the young adult group. Similarly, the dipeptide transported into cardiac sarcolemmal vesicles of the old hearts was the lowest compared with those from middle-aged and young adults [43]. In the kidney, the levels of rat PepT2 showed age-dependent expression [114]. PepT2 expression increased steadily during the first 14 days after birth [114]. A monotonic increase in PepT2 was observed in the expression from fetal d20 to d75 (adult) and was more pronounced in older groups [114]. This is consistent with Alghamdi et al. (2019) who identified that PepT2 was regulated by aging and that the expression of PepT2 in the kidney of aged rats was increased compared with both young adult and middle-aged rats, but no remarkable changes were observed between these two groups [113]. The expression of mouse PepT2 in the prostate was increased during development and remained at a plateau level after sexual maturation [38].

However, the regulation of PepT2 in the brain by aging is opposite to that of other organs. The expression of PepT2 in the cerebral cortex was highest in the fetus and decreased gradually in the 75 days after birth [17]. PepT2 was expressed in both astrocytes and neurons of neonatal rats, whereas it was not expressed in astrocytes of adult rats [17]. This finding is in accordance with a previous study showing that the expression of PepT2 in the cerebral cortex decreased with age [115]. PepT2 was developmentally expressed in the cerebral cortex, with the highest expression in the fetus, and it decreases rapidly with age [115]. The protein expression of PepT2 was highest in the fetal cerebral cortex, declining to 14% of the maximal level in adult rats [17]. In the lung, PepT2 was expressed in type II cells, but its expression was reduced during transdifferentiation and barely expressed in type I-like cells [40,116]. PepT2 was expressed in neurons (adult and neonate) and astrocytes (neonate but not adult) and showed an age-related decrease in the cerebral cortex. The cerebral cortex of the fetal and neonatal tissue has a higher expression than that of adults. The mRNA level of PepT2 was upregulated during the transition from pregnancy to lactation in the mammary glands of rats and humans, but it was reduced in pig mammary glands [23,24].

### 7.4. Regulation by Disease

The expression of PepT2 is regulated by pathological conditions [104]. As inflammation is a common feature of many diseases, the regulatory effect of inflammation on PepT2 is described in this review. Inflammation has been shown to regulate the expression and transport activity of many transporters, including PepT2, in mammals [93]. In the choroid plexus, PepT2 mRNA was enhanced due to peripheral inflammation [105,106]. Consistently, it was also revealed that inflammation affected the expression of PepT2 in the brain and kidney. Renal PepT2 expression was slightly inhibited, and renal clearance of cefadroxil mediated by PepT2 was substantially reduced in LPS-treated mice [106]. The mRNA expression of PepT2 was downregulated by TNFα and IFNγ in RWPE-1 cells, revealing that inflammation has an effect on the expression of PepT2 in prostate cells [38].

In addition to inflammation, hypertension can reduce the expression of PepT2 [43]. Y-SHR hypertrophied hearts in SHR rats (hypertensive young adult) showed a decline in the expression and transport activity of PepT2 compared with the WKY (control group).

### 7.5. Regulation by Posttranscriptional Modifications

A number of proteins in eukaryotic cells undergo posttranscriptional modifications. N-linked glycosylation acts as a mechanism for posttranslational modifications after protein synthesis and is proven to be crucial for the folding, stability, subunit assembly, localization and substrate binding of nutrient transporters [117,118,119]. N-linked glycosylation is commonly identified in transporters, such as hPepT2, bPepT2 and rPepT2 [27]. Human PepT2 (hPepT2) and bPepT2 were reported to contain five putative glycosylation sites and rPepT2 is composed of four N-linked glycosylation sites located on the large extracellular loop between α-helices 9 and 10 [10,27,69,71]. It has been shown to critically influence the transport activity of PepT2 [27]. Wang et al. (2020) revealed that bPepT2 is N-glycosylated in the great extracellular loop between the 9 and 10 TMDs, and lower transport activity was observed in the mutant group in which all of the N-glycosylation sites of PepT2 were mutated [27].

Neuronal precursor cell-expressed developmentally downregulated 4 isoform 2 (Nedd4-2) has been observed to interact with many transporters [120,121]. Wang et al. (2022) demonstrated that PepT2 is modulated by the mTOR signaling pathway, and Nedd4-2 mediates the regulation of PepT2 by mTORC1 [121]. USP18 (Ubiquitin-like specific protease 18) is an enzyme that cleaves ubiquitin from target proteins and modifies the transport activity of PepT2 [120]. Warsi et al. (2015) found that USP18 was able to enhance Vmax, which is upregulating the amount of plasma membrane expression of PepT2 expressed in Xenopus laevis oocytes [120].

### 7.6. Regulation by Other Proteins

Several factors can regulate the expression and transport activity of PepT2. For example, PDZ domain-containing protein (PDZK1) is regarded as an orchestrating scaffold to gain synergistic functions [122]. In 2005, the posttranscriptional regulation of PepT2 by PDZK1 was first identified. The PepT2–PDZK1 interaction was identified to function in the transport of di- and tripeptides and transportation of peptidomimetic drugs in the human kidney [108,109]. The capacity of PDZK1 to couple PepT2 to the Na+/H+ exchanger NHE3 is essential for producing the lumen-to-cell proton gradient [108]. Protein kinase C (PKC) was discovered to deliver the influence of [Ca^2+^] to peptide transport [110,123]. Consistently, it has also been proven that activation of signaling pathways that include PKC affects the kinetic properties of PepT2 in the LLC-PK1 [110].

The phosphatidylinositol-3-kinase (PI3K)/protein kinase B (Akt) pathway can modify cell growth, survival and nutrient transport [26,31,124,125]. Research has proven that the transport of some amino acids and small peptides in cells is regulated by PI3K [26,31]. Wang et al. (2019) revealed that PI3K and Akt inhibition significantly decreased the expression of PepT2 and the transport of model dipeptides and also found that the transportation of peptides in BMECs is controlled by the PI3K−Akt signaling pathway [26]. Takano et al. (2019) revealed that p38 MAPK and the nuclear factor κB pathway could regulate the innate immune response caused by bacterial peptides transported by PepT2 in H441 cells [62]. Numerous studies have proven that the expression and function of PepT2 are controlled by multiple factors, but it remains uncertain how intracellular signaling events proceeds after multiple stimulations and which transcription factors are involved in the regulation of PepT2.

## 8. Conclusions

In recent years, significant progress has been made in clarifying the effect of PepT2 in physiopathology in mammals, providing innovative insights into its potential pharmacological effect. PepT2 acts in peptide uptake and drug pharmacokinetics and has been targeted to improve the oral bioavailability and renal retention of many drugs. Since the current knowledge of PepT2 has led to some initial success in drug development, improving our knowledge of PepT2 is important and future studies need to address its 3D structure, protein–protein interactions, regulation by miRNAs and transcription factors and role as a cellular transporter. A deep understanding of the function of PepT2 can better utilize its biological properties for clinical treatments.

## Figures and Tables

**Figure 1 cells-11-02874-f001:**
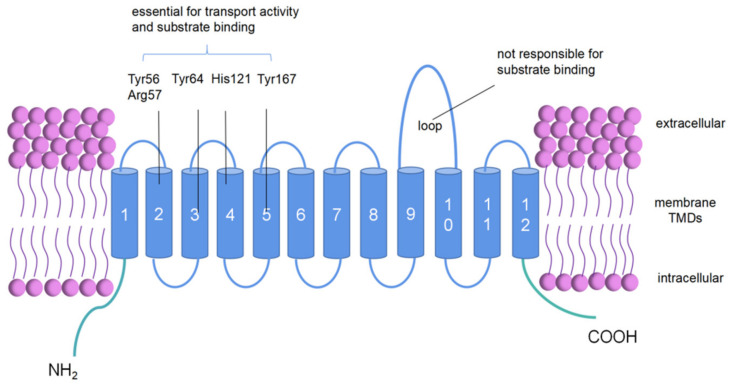
Topological structure of human PepT2 (hPepT2).

**Figure 2 cells-11-02874-f002:**
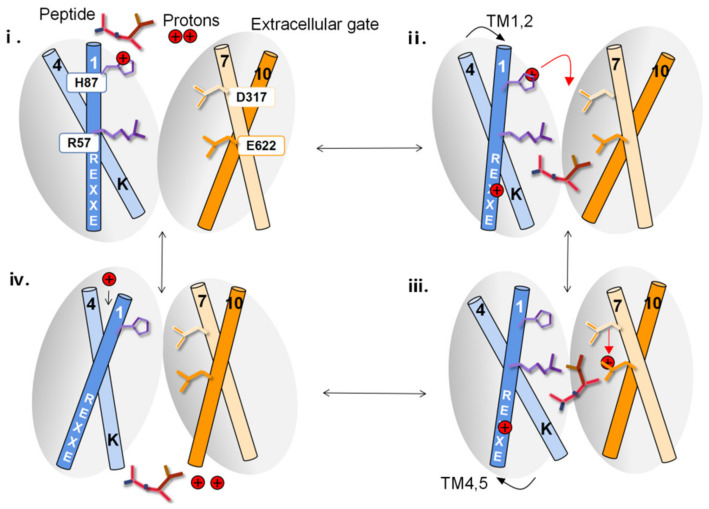
Transport mechanism of PepT2. Schematic representation of the transport cycle for PepT2 has been described in the text.

**Table 1 cells-11-02874-t001:** Localization and function of PepT2.

Tissue	Localization	Species	Function	References
Brain	the apical membrane of epithelial cells of the choroid plexus	Human, Rat, Mouse	responsible for the uptake of carnosine from cerebrospinal fluid	[17,19,20,36,37];
Reproductive organ	reproductive organs including testis, prostate, ovary and uterus	Rat, Mouse	absorption of di- and tri-peptides and peptidomimetic drugs in reproductive organ	[9,38]
Spleen	splenic lymphocytes and macrophages	Mouse, Human	mediate the innate immune response to bacterially produced chemotactic peptidomimetics in the spleen, especially macrophages	[31]
Kidney	brush border membrane of the epithelial cells in the proximal tubule	Human, Rat, Mouse, Rabbit	tubular reabsorption of carnosine into epithelial cells	[10,15,16,39]
Lung	bronchial epithelial cells and alveolar type II epithelial cells	Human	mediates the active translocation of peptides across the lung	[40]
Mammary gland	epithelial cells of mammary glands and ducts	Rat, Rabbit, Bovine, Human, Pig, Caprine	mediates the transport of peptides into the mammary gland	[23,25,26,27,41,42]
Skin	keratinocytes	Human	mediates the absorption of di-peptide, tripeptides and peptidomic drugs	[29]
heart	cardiomyocytes	Pig, Rat	serves to peptide uptake	[43,44]

**Table 2 cells-11-02874-t002:** Factors regulate the gene and or protein expression and function of PepT2.

Factors	Effect on PepT2	Modulation Type	References
Substrates, hormone			
Met-Met dipeptide;threonyl-phenylalanyl-phenylalanine tripeptide	increased the protein expression of PepT2 in BMEC	translational, functional	[101,102]
Protein restriction	decreased the mRNA expression levels of PepT2 in the kidney of nursery pig	transcriptional	[38]
Prolactin, Hydrocortisone, Insulin	enhanced the protein expression of PepT2 in bovine mammary gland	transcriptional	[102]
Epidermal growth factor	decreased in the transport capacity and expression of PepT2 in SKPT cells; enhanced peptide transport activity of PepT2 in LLC-PK1 cells	translational, functional	[35,103]
Thyroid	downregulated the mRNA expression of PepT2 in male rat kidney	transcriptional	[9]
Corticosteroids	inhibited the function of PEPT2 and resulted in lower bacterial peptide-induced into alveolar epithelial cells	functional	[104]
**Disease**			
Peripheral inflammation	PepT2 mRNA was increased due to peripheral inflammation	transcriptional	[105,106]
LPS-treated acute inflammation	Renal PepT2 expression was slightly inhibited in LPS-treated mice	transcriptional, functional	[107]
Prostate inflammation	the mRNA expression of PepT2 was downregulated by TNFα and IFNγ in RWPE-1 cell	transcriptional	[38]
Hypertension	reduced the expression of PepT2	translational	[43]
**other protein**			
PDZK1	PepT2—PDZK1 interaction was identified to function in the transport of di- and tri-peptides as well as peptide-like drug transport in the human kidney	translational, functional	[108,109]
PKC	affected the kinetic properties of PepT2 in renal cell lines LLC-PK1	functional	[110]
PI3K	regulated the xpression of PepT2 and the transport of model dipeptides	transcriptional, functional	[26]
MAPK	regulated the expression of PepT2 uptake of bacterial peptides in H441 cells	functional	[62]
NFκB	regulated the expression of PepT2 uptake of bacterial peptides in H442 cells	functional	[62]

## Data Availability

Data sharing not applicable to this article because no datasets were generated or analyzed during the current study.

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
