# Peer review of "Biology of Peptide Transporter 2 in Mammals: New Insights into Its Function, Structure and Regulation"

_cells, 2022, doi:10.3390/cells11182874_

Round 1
Reviewer 1 Report
This review manuscript is devoted to the peptide transporter PepT2 with emphasis on its structure, substrate specificity, tissue localization, regulatory factors, including pathological states. The review is timely, generally well written, and covers most important publication. I have only very minor remarks.
Line 29: should be “proton electrochemical POTENTIAL gradient”.
Lines 53-54: the sentence is not quite clear.
Line 86: “may” would be more appropriate here.
Line 149: “Figure 1” may be more appropriate after “helices” in this sentence as the figure does not show folds.
Lines 244-250: please explain to what two numbers of the amino acid residues refer.
Titles of section 7.2 and thereafter: Please consider using “regulation” instead of “regulated”, like in section 7.1.
There are language problems throughout the manuscript. I would recommend to consider sending the manuscript to an English editing company.
Author Response
Response to Reviewer 1
This review manuscript is devoted to the peptide transporter PepT2 with emphasis on itsstructure, substrate specificity, tissue localization, regulatory factors, including pathological states. The review is timely, generally well written, and covers most important publication. I have only very minor remarks.
AU: Thanks for your suggestion. We have tried our best to make corrections and revised our manuscript according to our suggestions.
Line 29: should be “proton electrochemical POTENTIAL gradient”.
AU: Thanks for your suggestion. We have adjusted into “proton electrochemical potential gradient”.
Lines 53-54: the sentence is not quite clear.
AU: We have adjusted into “In the kidney, PepT2 is predominantly expressed in the latter sections of the proximal tubule”.
Line 86: “may” would be more appropriate here.
AU: Revised as you suggested. “may” has been added into the manuscript.
Line 149: “Figure 1” may be more appropriate after “helices” in this sentence as the figure does not show folds.
AU: Revised as you suggested. “Figure 1” has been removed to after “helices”.
Lines 244-250: please explain to what two numbers of the amino acid residues refer.
AU: The numbers in the front represents the amino acid residues of PepT2 in Eukaryotes, prokaryotes in parentheses. As in this manuscript, we discuss the biology of PepT2 in mammals, so we delete the amino acid residues in parentheses.
Titles of section 7.2 and thereafter: Please consider using “regulation” instead of “regulated”, like in section 7.1.
AU: Revised as you suggested. “regulation” has replaced “regulated” in section 7.2-7.6.
There are language problems throughout the manuscript. I would recommend to consider sending the manuscript to an English editing company.
AU: Revised as you suggested. The language problems of this manuscript has been modified by an English editing company.
Reviewer 2 Report
The review entitled “Biology of peptide transporter 2 in mammals: New insights into its function, structure, and regulation” by Wang et al., represents an up-to-date, comprehensive, and highly interesting review of both the pathophysiological and pharmacological role of the PepT2 transporter. The great relevance that this transporter has been acquiring in recent years, both physiologically and pharmacologically, makes it a protein of great interest. Moreover, the recent resolution of the structure of the transporter, together with computational analyses, as well as the existing functional information, will allow a much more precise knowledge of the mechanisms related to the recognition and translocation of substrates and drugs, essential information for the intelligent design of pharmacologically active molecules. In summary, this comprehensive review is of great interest to all researchers interested in the PepT2 transporter.
However, I feel that the writing should be improved quite significantly, as there are many grammatical errors, as well as a haphazard use of verb tenses during the writing of the paper. In addition, the lack of connectors between sentences makes the reading not very agile. Therefore, I suggest a revision by a qualified English grammar editor, which would allow for a smooth and pleasant reading of the review.
Anyway, apart from the aforementioned English grammar revision, a list of suggested changes is also included:
1) Please check the spaces between text and references throughout the text. For example in line 41 (“delivery[12-14].”) and Table 1 (e.g. [17]; [19] in line 1 vs [16];[15] in line 6).
2) Line 50: An extra space between “was” and “mainly” should be removed.
3) Lines 53-55. High affinity and low capacity transporters are normally located in the latter section of the proximal tubule, while low-affinity high capacity ones are located in the S1 section. It has a physiological meaning that would be nice to discuss in this part of the review because connects function mechanism with physiology.
4) Lines 48-72. The lack of connectors between sentences results in excessive use of “PepT2”, throughout the paragraph.
5) Lines 69-72. Remove this paragraph as this information is repeated in the Table and at the beginning of the first paragraph in section 2. Just add a sentence like “Table 1 summarizes PepT2 tissue distribution in mammals” or something similar.
6) Table 1. Homogenize the use of capital letters. E.g. “Human” or “Brain” vs “located” or “responsible” in the first line of Table 1.
7) Table 1. In the “Localization” column, avoid using “located at” or “expressed in”, just add the localization. In this way authors will reduce the text in Table 1, easing its comprehension.
8) Line 107. Change “removel” by “removal”.
9) In lines 113, 131, or 134, publication data is appearing after the author (e.g. Wang et al. (2019)), while in lines 49 and 89, for example, the year is not indicated. Please homogenize the citing style.
10) Lines 127-135. The text needs to be reorganized. While the rest of Section 3 is quite organized, the text in lines 127 to 135 includes information regarding PepT2 in the mammary gland, neurons, and keratinocytes in a not very well-ordered way.
11) Structure of PepT2, Transport mechanism, and Substrate recognition sections should be reorganized for a better understanding. Mechanisms of substrate recognition and translocation are often difficult to clearly state in a comprehensive way for a wide variety of readers. Thus, structural information should be written in a very clear way, indicating only the most important information, essential for the understanding of the review.
12) Structure of PepT2. Figure 1 should be removed and substituted with a Figure showing the PepT2 fold. It has no sense to talk about the MFS fold showing a topological model. Additionally, the PepT2 structure also shows TMs and N- and C-t localization. Moreover, outward and inward-facing states of mammalian PepT2 in the presence and absence of substrate are available, opening the door to the identification of specific residues responsible for differential substrate recognition depending on the transporter side.
13) Structure of PepT2. Information regarding glycosylation or phosphorylation sites should be removed from here and included in section 7.5.
14) Structure of PepT2. A Figure showing the PepT2 binding site in the presence of a substrate, indicating essential residues for substrate recognition, and indicating the localization of previously identified important residues (e.g. R57, H121, Y56, etc…) will help in the understanding of this part of the review.
15) Line 169: what is bPepT2? Bovine?.
16) Line 177: Shewanella oneidensis should be in italics.
17) Line 183. Refer also to Parker et al., 2021 (rPepT2 structure).
18) Lines 192- 200. This part is the transport mechanism and should be removed from here and included in the Transport mechanism section.
19) To better understand the molecular mechanisms of substrate translocation, I would indicate first PepT2 substrate recognition mechanisms. So, after the Structure of PepT2, where a clear Figure indicating residues responsible for substrate recognition, Substrates of PepT2 section should be included, making the story more linear.
20) Substrates of PepT2. Check Parker et al., 2021 to better understand molecular mechanisms responsible for high and low-affinity substrate recognition.
21) Transport mechanism. Include a Figure with a sequence alignment, including prokaryotic and eukaryotic PepT2. Indicate residues mentioned in the text.
22) Line 195: What does His61(57) mean? I suppose it refers to residue numbering in different species. If so, please add this information regarding which species numbering are authors referring to.
23) Lines 224-240. Homogenize residue nomenclature. Line 224 “His87”, line 227 “His 87” and “Glu622”. Homogenize throughout the text.
24) Lines 244-250. Should be included in Substrates of PepT2 section.
25) Table 2. Same comments as those for Table 1.
26) Lines 298 and 301. Homogenize [14C] glycilsarcosine writing.
27) Lines 361 and 379. Change “Regulated” by “Regulation”.
28) Line 387. Change [Ca2+] by [Ca2+].
29) Line 398. Add a space between “[62].”and “Numerous”.
Author Response
Response to Reviewer 2
The review entitled “Biology of peptide transporter 2 in mammals: New insights into its function, structure, and regulation” by Wang et al., represents an up-to-date, comprehensive, and highly interesting review of both the pathophysiological and pharmacological role of the PepT2 transporter. The great relevance that this transporter has been acquiring in recent years, both physiologically and pharmacologically, makes it a protein of great interest. Moreover, the recent resolution of the structure of the transporter, together with computational analyses, as well as the existing functional information, will allow a much more precise knowledge of the mechanisms related to the recognition and translocation of substrates and drugs, essential information for the intelligent design of pharmacologically active molecules. In summary, this comprehensive review is of great interest to all researchers interested in the PepT2 transporter. However, I feel that the writing should be improved quite significantly, as there are many grammatical errors, as well as a haphazard use of verb tenses during the writing of the paper. In addition, the lack of connectors between sentences makes the reading not very agile. Therefore, I suggest a revision by a qualified English grammar editor, which would allow for a smooth and pleasant reading of the review.
AU: Thanks for your suggestion. We have tried our best to make corrections and revised our manuscript according to our suggestions. The language problems of this manuscript has been modified by an English editing company.
Anyway, apart from the aforementioned English grammar revision, a list of suggested changes is also included:
- Please check the spaces between text and references throughout the text. For example in line 41 (“delivery[12-14].”) and Table 1 (e.g. [17]; [19] in line 1 vs [16];[15] in line 6).
AU: Thanks for your suggestion. Revised as you suggested.
- Line 50: An extra space between “was” and “mainly” should be removed.
AU: Revised as you suggested.
- Lines 53-55. High affinity and low capacity transporters are normally located in the latter section of the proximal tubule, while low-affinity high capacity ones are located in the S1 section. It has a physiological meaning that would be nice to discuss in this part of the review because connects function mechanism with physiology.
AU: Thanks for your suggestion. This information “The distribution of peptide transporters is heterogeneous in kidney. Peptides and peptidomimetic drugs are absorbed sequentially in kidney, first by PepT1 and second by PepT2.” has been added into the manuscript.
- Lines 48-72. The lack of connectors between sentences results in excessive use of “PepT2”, throughout the paragraph.
AU: Thanks for your suggestion. We have added more connectors to avoid excessive use of “PepT2”.
- Lines 69-72. Remove this paragraph as this information is repeated in the Table and at the beginning of the first paragraph in section 2. Just add a sentence like “Table 1 summarizes PepT2 tissue distribution in mammals” or something similar.
AU: Revised as you suggested.
- Table 1. Homogenize the use of capital letters. E.g. “Human” or “Brain” vs “located” or “responsible” in the first line of Table 1.
AU: Revised as you suggested.
- Table 1. In the “Localization” column, avoid using “located at” or “expressed in”, just add the localization. In this way authors will reduce the text in Table 1, easing its comprehension.
AU: Revised as you suggested.
- Line 107. Change “removel” by “removal”.
AU: Revised as you suggested.
- In lines 113, 131, or 134, publication data is appearing after the author (e.g. Wang et al. (2019)), while in lines 49 and 89, for example, the year is not indicated. Please homogenize the citing style.
AU: Revised as you suggested.
- Lines 127-135. The text needs to be reorganized. While the rest of Section 3 is quite organized, the text in lines 127 to 135 includes information regarding PepT2 in the mammary gland, neurons, and keratinocytes in a not very well-ordered way.
AU: Thanks for your suggestion. “PepT2 is mainly located in the kidney, brain, and lung, recently it is also found to be located in mammary gland, neurons, and keratinocytes”. This has been added into the manuscript to be more logical.
- Structure of PepT2, Transport mechanism, and Substrate recognition sections should be reorganized for a better understanding. Mechanisms of substrate recognition and translocation are often difficult to clearly state in a comprehensive way for a wide variety of readers. Thus, structural information should be written in a very clear way, indicating only the most important information, essential for the understanding of the review.
AU: Revised as you suggested. Structural information of PepT2 has been rewritten to be more clear for the readers.
- Structure of PepT2. Figure 1 should be removed and substituted with a Figure showing the PepT2 fold. It has no sense to talk about the MFS fold showing a topological model. Additionally, the PepT2 structure also shows TMs and N- and C-t localization. Moreover, outward and inward-facing states of mammalian PepT2 in the presence and absence of substrate are available, opening the door to the identification of specific residues responsible for differential substrate recognition depending on the transporter side.
AU: Thanks for your suggestion. We have rewritten structure of PepT2 to be more clear. We also have revised to correct the error that using Figure 1 to show the PepT2 fold. Because the Figure showing the PepT2 fold is very difficult to draw and we try to cite the figure showing the structure of PepT2 published before, but it is a great pity that has not been authorized.
- Structure of PepT2. Information regarding glycosylation or phosphorylation sites should be removed from here and included in section 7.5.
AU:Revised as you suggested. Information regarding glycosylation or phosphorylation sites should be removed from Structure of PepT2 and included in section 7.5.
- Structure of PepT2. A Figure showing the PepT2 binding site in the presence of a substrate, indicating essential residues for substrate recognition, and indicating the localization of previously identified important residues (e.g. R57, H121, Y56, etc…) will help in the understanding of this part of the review.
AU: Thanks for your suggestion. The binding site of PepT2 has been shown in the topological structure of PepT2 (figure 1).
- Line 169: what is bPepT2? Bovine?
AU: bPepT2 represents Bovine PepT2. This information has been added into the manuscript.
- Line 177: Shewanella oneidensis should be in italics.
AU: Revised as you suggested.
- Line 183. Refer also to Parker et al., 2021 (rPepT2 structure).
AU: Revised as you suggested.
- Lines 192- 200. This part is the transport mechanism and should be removed from here and included in the Transport mechanism section.
AU: Thanks for your suggestion. Revised as you suggested.
- To better understand the molecular mechanisms of substrate translocation, I would indicate first PepT2 substrate recognition mechanisms. So, after the Structure of PepT2, where a clear Figure indicating residues responsible for substrate recognition, Substrates of PepT2 section should be included, making the story more linear.
AU: Revised as you suggested.
- Substrates of PepT2. Check Parker et al., 2021 to better understand molecular mechanisms responsible for high and low-affinity substrate recognition.
AU: Revised as you suggested.
- Transport mechanism. Include a Figure with a sequence alignment, including prokaryotic and eukaryotic PepT2. Indicate residues mentioned in the text.
AU: Thanks for your suggestion. As this manuscript is about the biology of peptide transporter 2 in mammals, so we deleted the information about prokaryotic PepT2. So we did not add the figure.
- Line 195: What does His61(57) mean? I suppose it refers to residue numbering in different species. If so, please add this information regarding which species numbering are authors referring to.
AU: It is a mistake. It is His61. We have adjusted this information in our manuscript.
- Lines 224-240. Homogenize residue nomenclature. Line 224 “His87”, line 227 “His 87” and “Glu622”. Homogenize throughout the text.
AU: Revised as you suggested.
- Lines 244-250. Should be included in Substrates of PepT2 section.
AU: Revised as you suggested.
- Table 2. Same comments as those for Table 1.
AU: Revised as you suggested.
- Lines 298 and 301. Homogenize [14C] glycilsarcosine writing.
AU: Revised as you suggested.
- Lines 361 and 379. Change “Regulated” by “Regulation”.
AU: Revised as you suggested.
- Line 387. Change [Ca2+] by [Ca2+].
AU: Revised as you suggested.
- Line 398. Add a space between “[62].”and “Numerous”.
AU: Revised as you suggested.